# Recent Advances in Syndactyly: Basis, Current Status and Future Perspectives

**DOI:** 10.3390/genes13050771

**Published:** 2022-04-27

**Authors:** Tahir Zaib, Hibba Rashid, Hanif Khan, Xiaoling Zhou, Pingnan Sun

**Affiliations:** 1Stem Cell Research Center, Shantou University Medical College, Shantou 515041, China; zaibi_hb@yahoo.com (T.Z.); xlzhou@stu.edu.cn (X.Z.); 2Guangdong Provincial Key Laboratory of Infectious Diseases and Molecular Immunopathology, Shantou University Medical College, Shantou 515041, China; 3Department of Biotechnology and Microbiology, Abasyn University, Peshawar 25000, Pakistan; hibbarashid9929@gmail.com; 4Department of Pharmacology, Shantou University Medical College, Shantou 515041, China; haneefsafi1@gmail.com

**Keywords:** syndactyly, heterogeneity, incomplete penetrance, surgery, *HOXD13*

## Abstract

A comprehensive summary of recent knowledge in syndactyly (SD) is important for understanding the genetic etiology of SD and disease management. Thus, this review article provides background information on SD, as well as insights into phenotypic and genetic heterogeneity, newly identified gene mutations in various SD types, the role of *HOXD13* in limb deformities, and recently introduced modern surgical techniques for SD. This article also proposes a procedure for genetic analysis to obtain a clearer genotype–phenotype correlation for SD in the future. We briefly describe the classification of non-syndromic SD based on variable phenotypes to explain different phenotypic features and mutations in the various genes responsible for the pathogenesis of different types of SD. We describe how different types of mutation in *HOXD13* cause various types of SD, and how a mutation in *HOXD13* could affect its interaction with other genes, which may be one of the reasons behind the differential phenotypes and incomplete penetrance. Furthermore, we also discuss some recently introduced modern surgical techniques, such as free skin grafting, improved flap techniques, and dermal fat grafting in combination with the Z-method incision, which have been successfully practiced clinically with no post-operative complications.

## 1. Background

Syndactyly (SD) is a congenital digital malformation characterized by webbing of the fingers and toes. Syndactyly is derived from the Greek word “syn”, meaning together, and “dactylos”, meaning digits. It is one of the most common hereditary limb disorders, with a prevalence of 3–10 in every 10,000 births, although higher estimates in the range of 10–40/10,000 have been reported [1,2,3,4]. Its occurrence in males is twice that in females, and mothers aged 40 years or older are more likely to produce children with inborn limb deformities compared to mothers who are 30 years of age or younger [5]. SD is genetic in origin; clinically, it is an extremely heterogeneous developmental deformity [6]. It may be symmetrical or asymmetrical and unilateral or bilateral. Moreover, inter- or intra-familial phenotypic variability is relatively common. The extent of variability of the disorder can even be observed in the same individual as he/she may have asymmetrical phenotypic features in the hands and feet, as well as between the right hand and left hand. SD can be completely or partially identified as bony or cutaneous, involving the phalanges, and may extend to the carpal and tarsal bones, even to the metacarpal and metatarsal levels of the limbs, and occasionally adjacent to the distal end of the forearm and foreleg.

Milder phenotypic features in limbs that are related to SD might be identified by interphalangeal fold differences [7]. SD mostly segregates as an isolated (non-syndromic) limb disorder but may occur in combination with other disorders (synostosis, acro-syndactyly, cleft hand, clinodactyly, polydactyly) or syndromes (Apert syndrome, Poland’s syndrome, Pfeiffer syndrome) [3]. Significant progress has been made in SD research, with multiple milestones being achieved during the past few years (Figure 1).

## 2. Classification of Syndactyly

The classification of non-syndromic SD can be performed in different ways, based on the various phenotypes. It can be simple or complex, complete or incomplete, and osseous (bone-involved) or cutaneous (only skin-involved). The classification system of Temtamy and McKusick for non-syndromic syndactyly is largely based on the phenotypic appearance (nature or site of affected limbs), along with segregation of the disorder in affected families [7]. The classification system of Temtamy and McKusick provided the basis for the latest modern classification system, which additionally considers advancements made clinically, as well as in basic molecular studies. In 2012, a nine-type classification system was put forward by Malik et al. that was mainly an extended version of the Temtamy and McKusick classification system [8]. The autosomal dominant mode of inheritance is evident in most of the types [9]. We summarize the classification of non-syndromic syndactyly in Table 1.

## 3. Variable Phenotypic Features of Non-Syndromic Syndactyly Types

The most-reported phenotypic features of SD are webbing of the 3rd and 4th fingers, while webbing of the 1st and 2nd digits rarely occurs because, during normal development, the thumb (1st digit) is not closely attached to the remaining fingers of the hand. There are two types of webbing. In the first type, only the skin is involved; this is referred to as simple syndactyly and can be sub-categorized into complete and partial SD; however, in the second type, the bones are also fused underneath the skin and this is called complex syndactyly [34]. 

In syndactyly type I (SD1), the clinical records show a large variation in patient phenotypes; this involves mesoaxial webbing, i.e., either the complete or partial blending of either the 3rd and 4th fingers, the 2nd and 3rd toe, or both in the same individual [4,10,11] (Figure 2A). A family with almost two dozen affected members showed deformities of digits that were characteristic of SD type 1c, with webbed fingers of both hands along with normal feet, but only one member of the family showed fused toes (3rd to 5th) [8,35].

Syndactyly and polydactyly (the addition of an extra digit in the limbs) may be found together in some cases. For example, syndactyly type II (SD2), also termed synpolydactyly (SPD), typically involves both webbing of the fingers and toe duplication, or an extra toe added in the feet [7], and usually involves webbing of the 3rd and 4th digits in the hands and the 4th and 5th toes, with an extra toe added [19]. The recognized phenotypic features of this type of SPD are webbing of the fingers (3rd and 4th) and toes (4th and 5th), which may be unilateral or bilateral and, rarely, evince duplicated toes and fingers [36] (Figure 2B–D). SPD is a very heterogeneous deformity among all SD types, in terms of phenotype as well as genotype [37]. The more complex type of SPD was first evident in a Belgium family with three affected family members who had irregular metacarpal as well as metatarsal synostoses [18]. 

The phenotypic features of syndactyly type III (SD3) are webbing of the 4th plus 5th fingers of both hands at the same time, but in some cases, the 3rd finger of each hand is also involved, accompanied by camptodactyly (Figure 2E) [21]. Similarly, syndactyly type IV (SD4) involves the webbing of the skin of all five digits of the hand, without the involvement of the bone; in most cases, polydactyly is also seen in the affected hands (Figure 2F). This type of SD is further classified into two categories, based on feet involvement along with affected hands. In the first category, no feet are involved, and in the second category, the fusion of the toes of one or both feet are involved [8,11]. Syndactyly type V (SD5) can be recognized by a bony combination of the 4th and 5th metacarpals in the forelimbs (Figure 2G) [27]. Additional deformities are also reported to be part of SD5, such as the irregular derivation of the fifth finger in both hands and unusual interphalangeal distortions [26]. Deformity of the feet involves defective metatarsals, for example, abnormal growth of the first metatarsal and the lesser small size of the remaining metatarsals, which severely affects the shape and function of the feet [8,26]. Correspondingly, syndactyly type VI (SD6) can be recognized by webbing of the four fingers (2nd to 5th) in the right hand, integration of the phalanges, and webbing of the 2nd and 3rd toes in the affected feet (Figure 2H) [7].

Syndactyly type VII (SD7) is the most severe form of SD, whereby the whole hand is distorted by the bony webbing of all fingers in the affected hand. The skeletal structure of the affected hand is fully disordered, to a degree that the phalanges cannot be distinguished as separate entities (Figure 2I) [38]. Furthermore, the carpals, metacarpals, and phalanges also have an uneven shape. Sometimes, other bones, such as the radius and ulna, also get affected, causing the length of the whole arm to shorten [7,23]. Two different phenotypic features of SD7 have been proposed, i.e., the spoon-head and oligodactyly types [39].

The main phenotypic feature of syndactyly type VIII (SD8) is the skeletal fusion of the 4th and 5th metacarpals, the shortness of the 4/5 metacarpals, and a few other small deformities in the skeletal structure of the affected hand (Figure 2J) [20]. Syndactyly type IX (SD9) can usually be recognized by phalangeal lessening, the osseous fusion of the metacarpals, 5th-finger clinodactyly, hypoplasia of the thumb and phalanges in the hand, and webbing of the toes (Figure 2K) [31,32]. 

## 4. Genetic Factors Underlying the Differential Phenotypes of Syndactyly

Mutations associated with the pathogenesis of SD have been recognized in numerous genes due to recent advancements in molecular genetics [40] (Figure 3). In 2000, SD1-b was mapped to chromosomal 2q34-q36 in the members of large families of German and Iranian kindred, but no specific gene has yet been identified [4,12,41], although recently a missense mutation (c.500A>G;p.Y167C) in the *HOXD13* gene has been reported to cause SD1-b [42]. Mutation in the *HOXD13*, present on chromosome 2, has also been reported to be associated with SD1-c. A study concerning two Chinese families affected with the Montagu type reported mutations in *HOXD13* [13]. Recently, missense variants (c.961A>C;p.T321P, c.917G>A;p.R306Q) in the *HOXD13* have been linked with SD1-c in families [42,43].

Different types of duplication, as well as missense and deletion variants in *HOXD13*, cause typical SPD disease [44]. Missense mutations in *HOXD13* have been linked with SPD1, which possibly affects the stability of the HOXD13 protein [45]. Recently, a missense variant (c.1157C>T;p.A375V) in the *TTC30B* has been reported in a Chinese family with SPD1 phenotypic features [46]. The *FBLN1* has been linked with SPD2, as mutations in this gene have been reported to result in a complex type of SPD [47]. The most common mutation involved polyalanine expansion or contraction in the N-terminal region of the HOXD13 protein [17].

Molecular evidence for SD3 has been confirmed in a family with SD3 and was linked to a locus at chr.7q36.3 [10]. Although SD3 is described in families as an isolated anomaly, it also occurs as a part of other diseases or syndromes [21,48,49]. 

Likewise, the duplication of 115.3 kb at a locus called ZRS (limb-specific cis regulator) on chromosome 7 has been linked with SD4 [22,23,24,25]. Recently, in two different studies, large duplications that involve several exons in the *LMBR1*, present on the same locus at chromosome 7, were associated with SD4 deformity in two large Chinese families [50,51].

A missense mutation (c.950A>G;p.Q317R) in the *HOXD13* has been confirmed to cause SD5 in a large Chinese family [27].

SD7 has been linked with the *LRP4* gene in several studies. For example, two brothers affected with SD7 deformity had a missense mutation (c.4910G>A;p.C1637T) in the *LRP4* [52]. In a large Pakistani family, a mutation in *L**RP4* (c.316+1G>A) has been reported to cause SD7 [53]. Similarly, another study reported a missense mutation (c.1151A>G;p.T384C) in the *LRP4* in a family affected with SD7 [54]. A deleterious variant (c.1348A>G;p.I450V) in *LRP4* was also associated with SD7 in two affected members of a Sri Lankan family [55]. 

In the case of SD8, a mutation in *FGF16* on the locus chrXq21.1 is the main cause, as two nonsense mutations (p.R179X and p.S157X) in *FGF16* have been linked with SD8 [56]. 

Likewise, a mutation in *BHLHA9* present on chromosome 17q13.3 has been linked with the SD9 [32]. Several other studies have reported missense (c.311T>C;p.I104T), frameshift (c.74delG;p.G25Afs*55), and deletion (c.252_270delinsGCA;p.F85Qfs*108) variants in the *BHLHA9* (Reference sequence: NM_001164405.2) in families affected with SD9 [33,57,58]. 

## 5. Some Excluded Types of Syndactyly and Underlying Genetic Factors

According to the current classification system, syndactyly can be classified into nine types, but this classification system does not consider numerous other syndromic and non-syndromic forms of SD. For a better understanding of the genetic factors behind all SD types, these excluded types of SD must be considered because several genes in combination are involved in limb development at the embryonic stage. For example, Saudi-type familial SD has been linked to the hammer-toe locus in mice [10], while Cenani–Lenz SD is associated with APC variations [59], missense alterations in *FIBULIN1* are associated with brain atrophy-syndactyly syndrome [60], and genomic replications of the SHH enhancer ZRS lead to triphalangeal thumb polysyndactyly syndrome [61], Greig syndrome, acrocephalosyndactyly syndromes and other SD phenotypes linked with the *GLI3* variants [62].

A large family affected with polydactyly and SD was shown to have a disease-linked variant (c.739C>T;p.Q247X) in the *GLI3* gene that was co-segregated in all affected family members [63]. Furthermore, a heterozygous mutation in the *NSDHL* (c.713C>A;p.T238N) gene has been reported in a nine-month-old female affected with a CHILD syndrome phenotype and SD, who has non-consanguineous parents [64]. In addition, the *TP63* gene has been found to be associated with SD in the presence of other abnormalities [65,66]. Recently, autosomal dominant SD has been associated with a microdeletion of 2.79 Mb at chr14q22-q22.2 in four affected members of a three-generation family with limb defects (syndactyly and polydactyly) along with other disorders, such as developmental delay and facial defects [67]. Recently, it has been reported that children with SD and prolonged heart-rate-corrected QT (QTc) interval have more multisystem diseases and electrocardiographic abnormalities [68]. Heterozygous missense alterations in *GLI3* (c.1622C>T;p.T541M) and *GJA1* (c.274T>C;p.Y92H) were identified in patients with the phenotypic features of SD type I [69], and two variants (p.D1403H, p.Q1564K) of *LRP4* have been reported in a child affected with isolated SD of both hands, although the *LRP4* gene has been reported to cause SD7 [70]. In recent studies, the combinations of SD, cleft hand, and polydactyly in a single patient suggested that some common genetic factors are behind these deformities [71,72]. Similarly, a missense variant (c.1622C>T;p.T541M) in *GLI3* has been reported in a patient with isolated postaxial synpolydactyly [73]. In another study, mutations in the *GJA1* gene, i.e., that located on chromosome 6q22-q23, have been reported to be linked with oculodentodigital dysplasia syndrome, and the SD3 phenotype has also been reported in some cases [74]. Recently, SD1-a has been reported to be associated with other diseases, e.g., diabetes [75].

We have listed the genes linked with these deformities in Table 2.

## 6. *HOXD13* and Its Role in Causing Syndactyly 

*HOXD13* belongs to a group of evolutionarily conserved *HOX* gene-family, which encode a group of transcription factors that regulate morphogenesis at an early embryonic stage [76]. Germline mutation in *HOXD13* is known to cause the deformity of limbs in humans. The phenomena of variable expressivity and incomplete penetrance are common with *HOXD13* mutations [77]. Mutations in *HOXD13* have also been linked with brachydactyly-syndactyly syndrome and VACTERL association [27,78]. *HOXD13* is known to cause different types of SD, e.g., SD1, SD5, and SPD1, which shows that the *HOXD13* gene has an important function in limb development (Table 3). The most common variation is a polyalanine expansion in the N-terminal domain of HOXD13, which is widely reported in families of different kindred (Figure 3B). For example, nine extra alanine residues that are added to the same region of HOXD13, due to the duplication of 27 bases, have been found in Turkish families with SPD1 [14]. In 2005, a polyalanine extension in *HOXD13* has been reported to cause SPD1 in four Danish families [79], while the duplication of 27 base pairs (c.184_210dup) has been reported to cause the addition of nine extra alanines to HOXD13 in a large Chinese family with SPD1 [80]. In 2009, a mutation within the N-terminal transcription-regulating domain of *HOXD13* (c.659G>T;p.G220V) was reported in a Greek family with a variant form of SPD [81]. In our own study, with the help of whole-genome sequencing (WGS), we identified a 24-base pair duplication variant, c.183_206dupGCGGCGGCTGCGGCGGCGGCGGC (Reference sequence: NM_000523.3) in *HOXD13* that results in the addition of eight extra alanines in four generations of a family in northern China [82]. Similarly, missense and nonsense mutations in *HOXD13* have also been reported in large families affected with SPD1 [83,84,85].

Previously, it has been reported that families inheriting a homozygous mutation in the *HOXD13* have a severe form of SPD [86] but, recently, this has been demonstrated not to be true in all cases of SPD1 patients with homozygous mutations [87]. Studies have also reported *HOXD13* mutations in families with different syndactyly types, e.g., SD1-a, SD1-c, and SD5 (Table 3), which gives a clear indication that *HOXD13* has a critical role in limb formation and that it may also interact with other limb-formation genes during the process. Recently, a study demonstrated how a missense mutation in the homeodomain of *HOXD13* leads to impaired transcriptional activity of *EPHA7* (one of the downstream genes of *HOXD13*) [88]. *EPHA7* is known to play a crucial role in limb development [89]. Hence, variations in the normal sequence of *HOXD13* can negatively affect other gene’s normal functions that could possibly results in differential phenotypes of SD.

**Table 3 genes-13-00771-t003:** List of *HOXD13* gene mutations reported for different types of non-syndromic syndactyly in the literature.

Mutation Type	cDNA Change	AA Change	NCBI Ref. Sequence	Allele	Phenotype	Ref.
Missense	c.917G>A	p.R306Q	NM_000523.4	Heterozygous	SD1-c	[13]
Missense	c.500A>G	p.Y167C	NM_000523.4	Heterozygous	SD1-b	[42]
Missense	c.961A>C	p.T321P	NM_000523.4	Heterozygous	SD1-c	[42]
Missense	c.917G>A	p.R306Q	NM_000523.3	Heterozygous	SD1-c	[43]
Duplication	c.183_206dup	p.A64_A71dup	NM_000523.3	Heterozygous	SPD1	[82]
Duplication	c.184_210dup	p.A63_A71dup	NM_000523.3	Heterozygous	SPD1	[80]
Duplication	c.183_206dup	p.A64_A71dup	NM_000523.4	Heterozygous	SPD1	[90]
Duplication	c.186-212dup	p.A63_A71dup	NM_000523.4	Heterozygous	SPD1	[91]
Missense	c.859C>T	p.G287X	NM_000523.3	Heterozygous	SPD1	[83]
Missense	c.556C>T	p.R186X	NM_000523.4	Heterozygous	SPD1	[84]
Missense	c.938C>G	p.T313R	NM_000523.4	Homozygous	SPD1	[85]
Missense	c.892C>T	p.R298W	NM_000523.2	Heterozygous	SPD1	[45]
Missense	c.659G>T	p.G220V	NM_000523.2	Heterozygous	SPD1	[81]
Missense	c.938C>G	p.T313R	NM_000523.3	Homozygous	SPD1	[86]
Missense	c.893G>A	p.A298G	NM_000523.3	Heterozygous	SPD1	[44]
Deletion	c.708delC	p.A236Lfs*30	NM_000523.4	Heterozygous	SPD1	[92]
Missense	c.925A>T	p.I309F	NM_000523.4	Heterozygous	SPD1	[88]
Splice donor site	c.781+1G>A	-	NC_000002.12 NM_000523.3	Heterozygous	SPD1	[93]
Missense	c.950A>G	p.Q317R	NM_000523.3	Heterozygous	SD5	[27]

## 7. Diagnosis and Surgical Treatment of Syndactyly 

SD is basically a limb malformation that belongs to congenital anomalies affecting bone or skeletal structure or function. It is caused when the digits of the fetus in the womb do not separate successfully, resulting in a webbed hand or feet. As an apparent deformity of the hands and feet, the characteristic is so obvious that it attracts instant attention or concern soon after birth, especially when it occurs in the hands. SD can be managed using different diagnostic tools, plus a genetic background of the patient’s family history and clinical data regarding deformities in affected family members. Genetic screening of the affected person and affected family members can easily reveal information about the genetic background, which can make it easy for a clinician to diagnose the deformity. Furthermore, other tools, such as X-rays and ultrasound, can also make the deformity clearer to the clinician and, therefore, more easily diagnosed. In the presence of all this information, a clinician will be able to diagnose the problem immediately and perform treatment effectively and efficiently [80,94]. Furthermore, after successfully diagnosing the deformity, genetic analysis of the patient and his family members will be helpful in establishing a clearer genotype–phenotype correlation. We have proposed a genetic analysis procedure in the form of a schematic diagram to obtain clearer genotype–phenotype correlation in future (Figure 4).

The most important aim and objective of surgical treatment for SD is to minimize possible complications, reinstate the space between the digits, and detach the limbs by using minimal medical techniques and avoiding problems that are likely to happen, such as recurrence and post-surgery complications, until a useful hand is obtained [95]. Skin implantation, open treatment, and the zigzag method of surgery techniques are usually conducted in corrective SD patient surgeries. Surgical outcomes in SD are more positive in simple-type SD compared to complex-type SD. In the United States, a recent study reported that the occurrence of SD is roughly 7 for every 10,000 babies born and that almost every affected child receives surgery before reaching two years of age. The study also pointed out that there may be some genuine problems in getting immediate health care, which includes accessibility to specialized surgeons for correcting limb deformity, failure to get to well-equipped hospitals, especially for people living in remote areas, and poor financial status [96].

The most significant and simple way to eliminate the deformity is early treatment of new-borns by surgery. For new-borns with simple SD, the best age to receive surgery ranges from 6 to 18 months old, whereas in case of complex syndactyly, surgery should be performed prior to 6 months of age [95]. It is very difficult to predict the effectiveness of the surgery because of the tremendous variety and phenotypic range of SD types. The simpler the SD, the higher the chance of achieving useful and fully recovered hand movement [97]. In case of simple SD, corrective and operational outcomes are typically excellent, with fewer chances of recurrence or the possibility of re-arising hand-related problems, whereas in case of complex SD, the chances of post-surgery complications are higher and involve difficulty in normal hand movement and nail deformities [23,97,98]. Complex-SD patients who have received surgery always require revisiting the clinician or surgeon to diagnose post-operative complications. 

Surgeons specializing in pediatrics often admit children with rare limb deformities. Closely associated deformities and syndromes should be always taken into consideration, because if not diagnosed accurately, surgery in that case can lead the patient into a worsen situation [99]. The main principle of the surgical treatment of SD and other associated limb deformities is to gain functional and useful limbs with less chance of recurrence. Skin grafts are the operational procedures most commonly used for corrective purposes in limbs affected with SD [100], although open-treatment methods for SD avoid leftover postoperative marks on the skin and are comparatively useful, with the best end results [101]. Several modern surgical techniques have been successfully practiced in the clinics with the aim of achieving useful limbs with no post-operative scars, smooth mobility of the digits, and fewer chances of recurrence. The free skin graft (full-thickness) surgical technique produces the best results when practiced in combination with the Z-method of incision, which can successfully diminish the scars usually obtained at the end of surgery and, as a result, can attain fully functional and useful limbs [102]. Recently, a technique called the improved flap technique was successfully implemented and involves the use of skin grafts with full thickness and different types of flaps to provide sufficient soft tissue cover. The results involved no post-operative complications, provided full recovery of the affected hand, and no discomfort to the child who received surgery [103]. Furthermore, the use of a dermal fat graft surgical technique specifically intended for treating the complex type of SD has recently been introduced [104]. In recent years, the use of abdominal flaps for complex SD release has also proved to be successful [105]. The part of the donor skin used for the corrective surgery of simple or complicated SD must have the features of both dorsal and palm skin as it can leave post-operative skin flaws in the digits, which can eventually affect mobility. Recently, it has been demonstrated that for the surgical purposes of SD, a gradation skin graft is far better cosmetically, compared to skin grafts from the sub-malleolar part, as it has been used traditionally; however, proper alignment with other parts is critical [106]. 

In case of the treatment of webbed toes in SD, it is slightly more complicated to perform surgery in children because of its high recurrence rate and post-surgery complications, especially in those children who are older than 24 months (the younger the age of the child at the time of surgery, lower the risk of recurrence) [107]. Recently, it has been demonstrated that the most effective surgical procedure suitable for both simple and complex SD involves the interdigitating of rectangular flaps because of its simple design, flexibility in alteration during surgery, and inclusive flap tips [108]. Another study suggested that the dorsal hexagon flap can be a useful substitute technique in treating syndactyly [109]. Furthermore, for skin grafting in which the donor site of the patient also gets disturbed, dorsal rectangular flaps can be very useful, with appealing results for the patient [110]. Based on the condition of the deformity, an individualized treatment plan should be made that can better restore the shape and function of the thumb, especially in SD5 [111]. Moreover, the use of methotrexate can reduce keloid formation just after the dissection of the webbed digits [112]. 

Post-operative check-ups of the patients should be frequently arranged prior to complete recovery to avoid any difficulties due to surgery [95]. Overall, treatment and surgery for SD are carried out by surgeons of other specialties, which shows that SD treatment is a harmless and effective process with few postoperative complications, but it does need to be followed up by clinicians to ensure fully recovered limbs [113]. 

## 8. Future Perspectives 

Due to modern techniques, more of the genetic factors behind SD are being revealed as research proceeds on inborn limb deformities. It is, therefore, of considerable importance to further elucidate the genetic etiologic factors that contribute to the differential phenotype and incomplete penetrance of all types of non-syndromic SD. Next-generation sequencing can play a crucial role in identifying new pathogenic genes and provide a better understanding of this deformity in the future [114]. More in vitro and in vivo studies should be conducted to investigate the interaction of *HOXD13* with other closely related genes that are involved in limb deformities. A stronger phenotype–genotype correlation needs to be established using the modern technologies of genetic engineering and biotechnology to investigate the factors involved in causing differential phenotype of SD.

## Figures and Tables

**Figure 1 genes-13-00771-f001:**
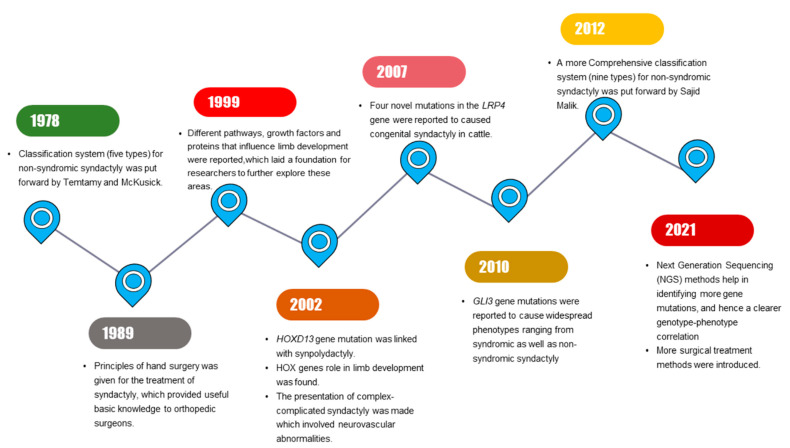
Schematic diagram illustrating the series of milestones achieved in past years in syndactyly research.

**Figure 2 genes-13-00771-f002:**
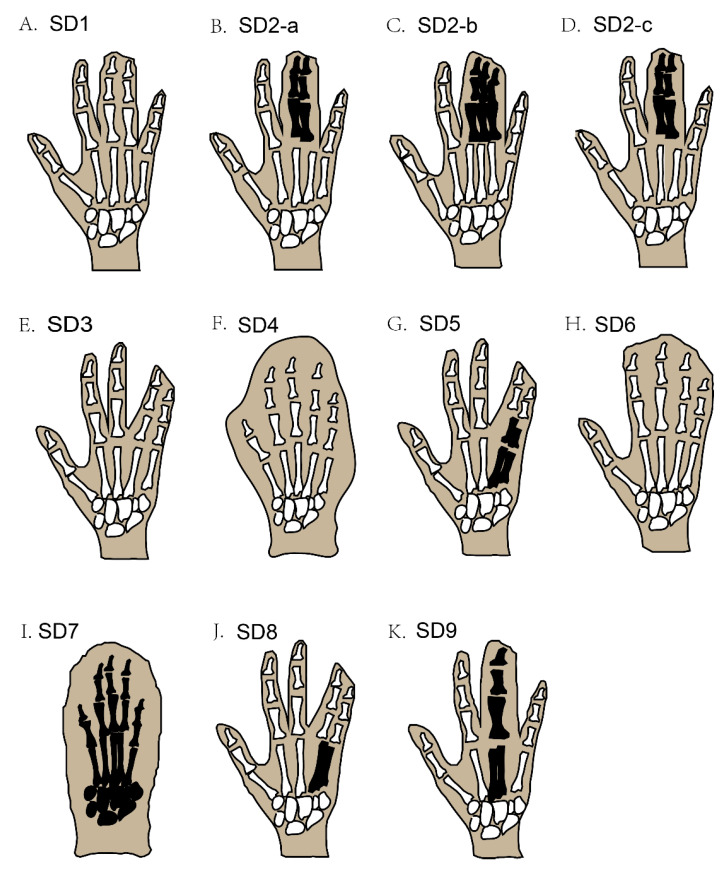
Schematic diagram of different types of non-syndromic syndactyly (types 1–9). Black-coloured areas represent the blending of the bones under webbed skin, while white-coloured areas under webbed skin represent unfused bones.

**Figure 3 genes-13-00771-f003:**
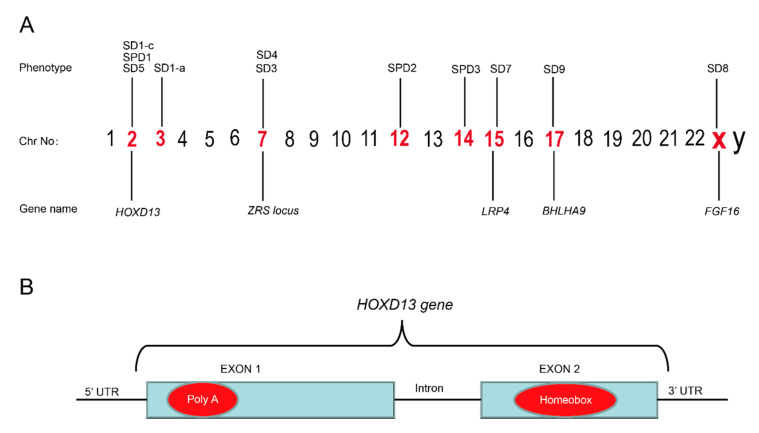
(**A**) Illustration of chromosomes and the syndactyly types belonging to different chromosomes. (**B**) The mutational hot spots in the poly(A) region in exon 1 and homeobox region in exon 2 of the HOXD13 gene are marked in red.

**Figure 4 genes-13-00771-f004:**
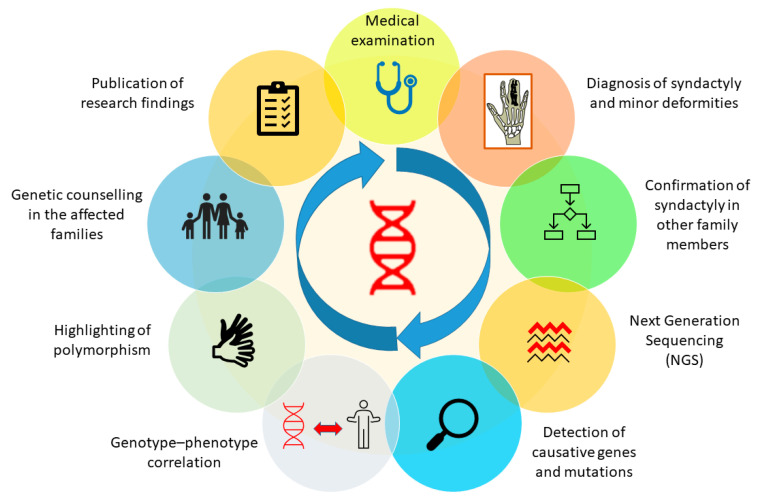
Genetic analysis procedure to attain a clearer genotype-phenotype correlation in all types of syndactyly.

**Table 1 genes-13-00771-t001:** List of genes and loci responsible for different types of non-syndromic syndactyly.

Clinical Phenotype	Original Name	Major Symptoms	Locus/Gene	Mutation Type	Inheritance	References
Syndactyly I-a	Zygodactyly	Cutaneous webbing of 2nd and 3rd toes without the hand involvement	Chr.3p21.31	-	AD*	[4,10,11]
Syndactyly I-b	Lueken type	Bilateral bony or cutaneous webbing of 3rd/4th fingers and 2nd/3rd toes	*HOXD13*	Duplication, missense, and deletion	AD	[4,12]
Syndactyly I-c	Montagu type	Bilateral bony or cutaneous webbing of 3rd/4th fingers, with normal feet	*HOXD13*	Duplication, missense, and deletion	AD	[4,13]
Syndactyly I-d	Castilla type	Bilateral cutaneous webbing of the 4th and 5th toes	-	-	AD	[4,8,14]
Syndactyly II-a	Vordingborg type	Distinct combinations of syndactyly and polydactyly	*HOXD13*	Duplication, missense, frameshift, splicing and deletion	AD	[15,16,17]
Syndactyly II-b		Metacarpal and metatarsal synostoses	*FBLN1*	Missense	AD	[15,18]
Syndactyly II-c		Cutaneous webbing, abnormal metacarpals	Chr.14q11.2-12	-	AD	[19]
Syndactyly III	Johnston-Kirby type	Bilateral complete syndactyly of the 4th and 5th fingers	Chr.7q36.3	-	AD	[10,20,21]
Syndactyly IV	Haas-type polysyndactyly	Complete cutaneous syndactyly of all fingers	*LMBR1*	Large duplications and missense	AD	[22,23,24,25]
Syndactyly V	Dowd type	Synostotic fusion of metacarpals	*HOXD13*	Duplication, missense, and deletion	AD	[1,26,27]
Syndactyly VI	Mitten type	Fusion of 2nd–5th fingers of the right hand	-	-	AD	[7]
Syndactyly VII-a	Cenani-Lenz syndactyly (CLS)	Bony fusion of all digits	*LRP4*	Missense	AR*	[28,29]
Syndactyly VII-b			15q13.3, *GREM1*-*FMN1*	-	-	[30]
Syndactyly VIII	Orel-Holmes type	Fusion of metacarpals 4/5	*FGF16*	Nonsense	XR*	[20]
Syndactyly IX	Mesoaxial synostotic syndactyly (MSSD)	Phalangeal reduction	*BHLHA9*	Missense, frameshift, and deletion	AR	[31,32,33]

AD* = autosomal dominant, AR* = autosomal recessive, XR* = X-linked recessive.

**Table 2 genes-13-00771-t002:** Genes linked with the excluded types of syndactyly.

Gene	Deformity/Syndrome	References
*APC*	Cenani-Lenz syndrome and other related syndactyly disorders	[59]
*FIBULIN1*	Atrophy-syndactyly syndrome	[60]
*GLI3*	Acrocephalo-syndactyly	[62]
*GLI3*	Polydactyly and syndactyly	[63]
*GLI3*	Isolated postaxial synpolydactyly	[73]
*NSDH*	CHILD syndrome phenotype and syndactyly	[64]
*TP63*	Syndactyly in combination with other abnormalities	[65,66]
*LRP4*	Isolated syndactyly	[70]
*GJA1*	Oculodentodigital dysplasia	[74]

## Data Availability

Not applicable.

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
