# Peer review of "Recent Advances in Syndactyly: Basis, Current Status and Future Perspectives"

_genes, 2022, doi:10.3390/genes13050771_

Round 1
Reviewer 1 Report
In this review, Zaib et al. describe current knowledge of non-syndromic syndactyly, a frequent malformation of the digits. The authors focus on the genetic basis of the different forms of syndactyly, and provide information about recent surgical procedures that are being implemented to correct these debilitating defects.
This comprehensive review is well written, up-to-date and well organized. The information provided in this work is expected to be useful for clinicians, geneticists and developmental biologists.
Author Response
Thank you so much to reviewer for the appreciating comments.
Reviewer 2 Report
Title: Recent advances in syndactyly: basis, current status and future 2 perspectives
Zaib et al.
This is an interesting review article on syndactyly. The authors have attempted to give updated review on the subject. The manuscript would, however, benefit from the following changes/amendments:
- The authors miss the feet phenotypes in figure 2. Without the feet phenotype you cannot diagnose the condition.
- Table 1. Mutation type. The data presented is incomplete as there are several mutation types in a certain gene in different syndactyly types. Please enlist all mutation type.
- Fig. 3A. Please replace LMBRI with ZRS locus.
- Table 2: It is not clear what is 'excluded syndactyly types'
- Table 3 is incomplete. There is an exhaustive list of expansion mutations reported for HOXD13 which need to be provided here. Without this list it is difficult to comprehend the mutation spectrum of HOXD13.
- Fig. 3.Please mention GJA1 in the diagram.
- The authors use arbitrary notation for the variants which is very confusing. For each variant, please follow the notation of ACMG classification.
- In the Abstract it is claimed that ‘It also highlights the series of events at a molecular level in humans that might result in simple or complex syndactyly’. However, this point is not found in the main text.
- In the Abstract the authors claim that ‘and how 23 HOXD13 interaction with other genes may be one of the reasons behind the differential phenotypes and incomplete penetrance’. However, this point is not discussed in the text.
- Page 6. Heading 4. Genetic factors underlying differential phenotypes of syndactyly types. The description of mutation is not complete. There are a number of mutations that have been skipped by the authors.
- It is not clear which is the updated information in this paper. Most of the stuff is just the repetition of previous review papers published on this subject.
- There are a number of typos and syntax errors which need to be corrected.
